# Examining and investigating the impact of demographic characteristics and chronic diseases on mortality of COVID-19: Retrospective study

**Abbas Al Mutair**[1,2,3], **Alya Al Mutairi**[ID][4]*, **Saad Alhumaid**[5], **Syed Maaz Abdullah**[6], **Abdul Rehman Zia Zaidi**[7], **Ali A. Rabaan**[8,9], **Awad Al-Omari**[7,10]

1 Research Center Director, Almoosa Specialist Hospital, Al-Ahsa, Saudi Arabia, 2 College of Nursing, Princess Norah Bint Abdulrahman University, Riyadh, Saudi Arabia, 3 School of Nursing, University of Wollongong, Wollongong, Australia, 4 Mathematics Department, Taibah University, Medina, Saudi Arabia, 5 Drug Information and Research Department, Administration of Pharmaceutical Care, Ministry of Health, Al-Ahsa, Saudi Arabia, 6 Department of Medicine, King Edward Medical University, Lahore, Pakistan, 7 Research Center, Dr. Sulaiman Al Habib Medical Group, Riyadh, Saudi Arabia, 8 Molecular Diagnostics Laboratory, Johns Hopkins Aramco Healthcare, Dhahran, Saudi Arabia, 9 Department of Public Health and Nutrition, The University of Haripur, Haripur, Pakistan, 10 College of Medicine, Alfaisal University, Riyadh, Saudi Arabia

* amutairi@taibahu.edu.sa

**Data Availability Statement:** All relevant data are within the manuscript and its Supporting information files.

## Abstract

### Background

Epidemiological features characterization of COVID-19 is highly important for developing and implementing effective control measures. In Saudi Arabia mortality rate varies between 0.6% to 1.26%. The purpose of the study was to investigate whether demographic characteristics (age and gender) and non-communicable diseases (Hypertension and Diabetes mellitus) have a significant association with mortality in COVID-19 patients.

### Methods

Prior to data collection, an expedite approval was obtained from Institutional Review Board (IRB Log No: RC. RC20.09.10) in Al Habib Research Center at Dr. Sulaiman Al-Habib Medical Group, Riyadh, Saudi Arabia. This is a retrospective design where we used descriptive and inferential analysis to analyse the data. Binary logistic regression was done to study the association between comorbidities and mortality of COVID-19.

### Results

43 (86%) of the male patients were non-survivors while 7 (14%) of the female patients were survivors. The odds of non-survivors among hypertensive patients are 3.56 times higher than those who are not having a history of Hypertension (HTN). The odds of non-survivors among diabetic patients are 5.17 times higher than those who are not having a history of Diabetes mellitus (DM). The odds of non-survivors are 2.77 times higher among those who

**Funding:** The authors received no specific funding for this work.

**Competing interests:** The authors have declared that no competing interests exist.

have a history of HTN and DM as compared to those who did not have a history of HTN and DM.

## Conclusions

Those patients that had a history of Hypertension and Diabetes had a higher probability of non-survival in contrast to those who did not have a history of Diabetes and hypertension. Further studies are required to study the association of comorbidities with COVID-19 and mortality.

## Introduction

Corona virus disease 2019 (COVID-19), which was first described in 2019, is caused by severe acute respiratory syndrome coronavirus 2 (SARS-CoV-2). Globally, as of April 6, 2021, there have been 131,020,967 confirmed cases of COVID-19; including 2,850,521 deaths, reported to the World Health Organization [1]. The majority of the cases were reported in Americas (43%), Europe (32%) and South-East Asia (14%). As of April 6, 2021, the total reported confirmed COVID-19 cases have reached more than 392,682 including more than 6,697 deaths within Saudi Arabia [2]. Human coronaviruses were first identified in the mid-1960s and usually cause mild upper-respiratory tract illness. On March 02, 2020, the first confirmed case of COVID-19 was reported from Saudi Arabia [3]. SARS-CoV-2 infection is associated with significant mortality related to the virulence of the virus, nature of the disease, and the lack of effective therapy. Patients with SARS-CoV-2 who develop acute respiratory distress syndrome (ARDS) are at a high risk of dying from refractory hypoxemia, multiorgan failure, and septic shock [4]. Studies shown up to 20% of the patients infected with SARS-CoV-2 develop high disease severity and need to be hospitalized [5, 6]. Intensive care unit (ICU) admission is a requirement for up to 26% among those who are hospitalized [7]. A consistent risk factor which has been singled out by the WHO and Centres for Disease and Control CDC is old age which might be the case for Saudi Arabia though the country has relatively young population [8]. Other risk factors associated with worse COVID-19 outcomes and high mortality rate include Diabetes mellitus, Hypertension and obesity [9]. Another unique prognostic factor in Saudi Arabia is the migrant labor workers who are mostly men coming from less developed countries and known of increasing prevalence of cardiometabolic diseases [10]. In the present study, we extracted retrospective data of 116 patients admitted in in-patient medical wards and intensive care units (ICUs) with laboratory-confirmed COVID-19 from different hospitals in Riyadh, Saudi Arabia through 20 April to 20 June 2020. However, the purpose of the study is to investigate whether demographic characteristics (age and gender) and non-communicable diseases (Hypertension (HTN) and Diabetes mellitus (DM)) have significant association with mortality in COVID-19 patients.

## Design

A retrospective design was employed in the current study where data were retrieved from a tertiary private hospital group in Saudi Arabia. Data were recruited from April 2020 to June 2020, data collection included reviewing patients' electronic medical records, nursing notes, laboratory characteristics, management details and clinical outcomes. A data collection form was prepared to obtain the needed COVID-19 patient information. Patients were de-identified

and no personal information were collected and therefore no informed consent was deemed necessary. In the present study the ethical guidelines of Declaration of Helsinki and good clinical practice was followed.

## Data analysis

IBM SPSS Statistics software, version 24.0 (IBM Corp., Armonk, NY, USA) was used to perform the current data analysis. In this study, descriptive and inferential analysis have been employed to analyse the data. Descriptive analysis such as frequency as a percentage has been utilized to calculate frequency and percentage for demographic variables and for comorbidities of the patients. Furthermore, binary logistic regression has been conducted to examine the association between (demographic variables and comorbidities) and mortality of COVID-19. Logistic regression is an extension of simple linear regression where the dependent variable is dichotomous or binary in nature.

## Results

### Demographic characteristics of COVID-19 patients

Demographic characteristics have been grouped under survivors (recovered) and non-survivors (died) as demonstrated in Table 1. There are two demographic variables which are age and gender. By looking at age (survivors), 21 (31.8%) of the total patients were at the age range 30–40 years old, 16 (24.2%) were 21–30 years old and above 50 years old, 11 (16.7%) were 41–50 years old and 2 (3%) were 10–20 years old. Meanwhile, for (non-survivors), over half of the patients which comprised 33 (66%) of the total respondents above 50 years old, 12 (24%) were 41–50 years old and 5 (10%) were 30–40 years old. In terms of gender, among survivors, 39 (59.1%) were females while 27 (40.9%) were males. It was reported that 43 (86%) of the male patients were non-survivors while 7 (14%) of the female patients were survivors. In addition, the results demonstrated that about 74 (63.8%) had no history of chronic diseases (HTN and DM). Meanwhile, 23 (19.8%) had one of the disease (HTN or DM), and 19 (16.4%) had a history of both HTN and DM.

**Table 1. Demographic characteristics of COVID-19 patients.**

| Demography Variables | Survivors | | non-survivors | |
|---|---|---|---|---|
| | n | % | n | % |
| **Age** | | | | |
| 10–20 Years Old | 2 | 3 | | |
| 21–30 Years Old | 16 | 24.2 | | |
| 30–40 Years Old | 21 | 31.8 | 5 | 10 |
| 41–50 Years Old | 11 | 16.7 | 12 | 24 |
| Above 50 Years Old | 16 | 24.2 | 33 | 66 |
| **Gender** | | | | |
| Male | 27 | 40.9 | 43 | 86 |
| Female | 39 | 59.1 | 7 | 14 |
| **Comorbidities** | | | | |
| No History HTN&DM | 53 | 80.3 | 21 | 42.0 |
| Having one of them (HTN OR DM) | 7 | 10.6 | 16 | 32.0 |
| Having both (HTN & DM) | 6 | 9.1 | 13 | 26.0 |

**Table 2. Binary logistic regression for demographic characteristics and mortality rate.**

| Variable Name | Adjusted OR | 95% Confidence Interval | | P Value |
|---|---|---|---|---|
| | | Lower | Upper | |
| Age | 1.069 | 1.036 | 1.103 | < 0.001 |
| Gender | 5.939 | 2.084 | 16.921 | 0.001 |

**Table 3. Binary logistic regression for chronic disease and mortality rate.**

| Variable Name | Adjusted OR | 95% Confidence Interval | | P Value |
|---|---|---|---|---|
| | | Lower | Upper | |
| HTN | 1.822 | 1.638 | 5.207 | 0.006 |
| DM | 4.011 | 1.523 | 10.561 | 0.005 |

## The association between demographic characteristics and mortality rate

Binary logistic regression has been computed to examine the association between demographic characteristics of the patients (age and gender) and mortality as demonstrated in Table 2. The findings demonstrated that age OR = 1.080, (95% CI 1.036–1.103) and gender OR = 5.930, (95% CI 2.084–16.920) have significant association with mortality. The odds of non-survivors are 1.069 times among older patients as compared to younger. According to gender, the odds of non-survivors among male patients are 2.084 times as compared to female patients. In other words, older patients more likely to die as compared to younger patients. Besides that, male patients more likely to die due to COVID-19 virus as compared to female patients.

## The association between history of chronic disease and mortality

Binary logistic regression was used to investigate the association between mortality and history of chronic disease (HTN and DM) as demonstrated in Table 3. The results of logistic regression showed that the history of HTN OR = 1.820, (95% CI 1.640–5.210) and DM OR = 4.011, (95% CI 1.523–10.561) have significant association with mortality. The odds of non-survivors among hypertensive patients are 1.822 times higher than among normotensive patients. Similarly, the odds of non-survivors among diabetic patients are 4.011 times higher than those who are not having a history of Diabetes.

Univariate logistic regression was computed to determine whether the history of a combination of HTN and DM may lead to higher mortality due to COVID-19 as demonstrated in Table 4. The findings demonstrated significant findings indicating that these factors contribute to mortality OR = 2.770, (95% CI 1.604–4.784). The odds of non-survivors are 2.770 times higher among those who have a history of HTN and DM as compared to those who did not have a history of HTN and DM.

**Table 4. Univariate logistic regression for combination HTN & DM with mortality rate.**

| Variable Name | Adjusted OR | 95% Confidence Interval | | P Value |
|---|---|---|---|---|
| | | Lower | Upper | |
| HTN & DM | 2.770 | 1.604 | 4.784 | < 0.001 |

## Discussion

Our study illustrates a significant relationship between variations in age and gender, prevalence of DM and HTN, and the overall COVID-19 related mortality. The results demonstrate that advanced age, male gender, and history of DM and HTN among COVID-19 patients are disproportionately prevalent among non-survivors and thus suggest that these mentioned demographic and disease elements are poor prognostic factors leading to an increased incidence of mortality. The binary logistic regression analysis used in our study quantitatively suggests that the COVID-19 patients who have age > 50 years, who are males, or who have history of HTN and/or DM may have up to 9-fold increased chances of mortality than those patients who are young, who are females and are non-diabetic and normotensive. This is in line with the previous reports that suggest similar correlation between mortality and these factors [11–18]. In a previously reported study conducted in Saudi Arabia HTN and type 2 Diabetes mellitus were the most comorbidities in both genders of all included patients accounting for 45.7% and 28.5% retrospectively [9] and this was expected as the present study deals with moderate to severe COVID-19 hospitalized patients. A meta-analysis conducted by [19] which looked over 1,994 COVID-19 patients exhibited an increased mortality among patients who were male and suggested a case fatality rate of 7% [18]. Another meta-analysis conducted by [20] (n = 6520) showed that Hypertension was linked to increased mortality among COVID-19 patients (RR 2.21 (1.74–2.81), p < 0.001) [19]. Although many of the studies involving the association of Hypertension with COVID-19 related mortality were confounded by other risk factors like age and gender, but in a meta-analysis (n = 15302) proved that Hypertension is an independent risk factor altogether and leads to an escalated risk of adverse outcomes including death in COVID-19 patients [19]. Previously reported studies which segregated the patients diagnosed with COVID-19 on the basis of age and showed that patients in the advanced age groups had proportionately increased case fatality rates reaching up to 20% in patients >80 years of age [21, 22]. This may also be the case in Saudi Arabia as high mortality and worse COVID-19 outcomes are associated with old age [8, 9]. This is crucial, keeping in view the progressive aging of the worldwide population as the number of older individuals aged 60 years or above is expected to increase to 2.1 billion by 2050 [23]. This means that the associated prognostic challenges associated with COVID-19 infection among elderly patients is expected to accrue with time. DM has also been proven to significantly exacerbate the prognostic outcomes among COVID-19 patients [24]. Study by [24] (n = 23698) showed that DM was associated with more than three-fold increased risk of in-hospital mortality in COVID-19 patients than in non-diabetics [25]. A study by [26] (2020, n = 72314) revealed a crude mortality rate of about 7.3% among COVID-19 patients who were Diabetics, in comparison to a rate of only 0.9% among non-diabetic COVID-19 patients [26]. Furthermore, in another retrospective case-control study, a mortality rate of around 35% was found in diabetic COVID-19 patients [27]. Locally in Saudi Arabia DM among COVID-19 patients has been observed to be associated with higher risk for sever outcomes including high mortality rate [9]. All these international and national studies reflect a strong impact of disease prevalence and demographic factors on the prognosis of COVID-19 infection. However, there have been studies that have shown that diseases like HTN and DM may have no association with increased mortality. In addition [28], (n = 209) showed that although HTN apparently seemed to increase the risk of mortality in univariate analysis (OR = 5.000, 95% CI [1.748–14.301]), but the multivariate analysis done in the same study proved otherwise and removed the possible association between HTN and mortality (OR = 1.099, 95% CI [0.264–4.580]) [28]. Another study by [29] (n = 257) showed the same pattern for both DM and HTN that the mortality association of both diseases disappeared when multivariate analysis was used in COVID-19 patients [29].

Study by [30] reported that septic shock and high neutrophil count were found to be predictive of death in severe COVID-19 patients.

The authors acknowledge several limitations aside from the retrospective design. First, it is possible that there was selection bias in this study as this is short-term cross-sectional retrospective observational study where data collected over short period of time as a prospective study would give us better insight into the study phenomena. Second, the follow-up was limited through June, 2020, hindering the possibility of including all outcomes. Consequently, there may have been some partiality regarding the prognosis of the patients. Thirdly, the results may be confounded with other concomitantly prevalent risk factors which were not taken into account like smoking, obesity, asthma, cardiovascular morbidity and related issues. Finally, some follow-up data were unavailable. Despite its limitations, this prospective study provides good data on 116 patients with COVID-19 who were treated in Saudi Arabia tertiary healthcare centres.

## Conclusion

The findings concluded that Demographic characteristics age and gender do have an impact on mortality rate of COVID-19. As reported that older patients more likely to non-survival as compared to younger patients. Besides that, it was recorded that male patients have higher tendency to non-survival as compared to female. In addition, those with history of Hypertension and Diabetes have higher probability to non-survival as compared with those who are not having history of Hypertension and Diabetes.

## Supporting information

**S1 Data.**
(SAV)

## Acknowledgments

The authors thank the referees for constructive comments.

## Author Contributions

**Conceptualization:** Abbas Al Mutair, Abdul Rehman Zia Zaidi.

**Data curation:** Saad Alhumaid, Syed Maaz Abdullah, Abdul Rehman Zia Zaidi, Ali A. Rabaan.

**Formal analysis:** Alya Al Mutairi.

**Funding acquisition:** Abbas Al Mutair, Awad Al-Omari.

**Investigation:** Alya Al Mutairi.

**Methodology:** Alya Al Mutairi.

**Project administration:** Abbas Al Mutair.

**Resources:** Saad Alhumaid, Syed Maaz Abdullah, Abdul Rehman Zia Zaidi, Ali A. Rabaan, Awad Al-Omari.

**Software:** Alya Al Mutairi.

**Supervision:** Awad Al-Omari.

**Validation:** Abbas Al Mutair, Syed Maaz Abdullah.

**Visualization:** Saad Alhumaid, Abdul Rehman Zia Zaidi.

**Writing – original draft:** Abbas Al Mutair, Saad Alhumaid, Syed Maaz Abdullah, Ali A. Rabaan.

**Writing – review & editing:** Abbas Al Mutair, Abdul Rehman Zia Zaidi.

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
