## [Decision Letter · Decision Letter 0]

30 Mar 2021

PONE-D-21-01309

Examining and Investigating the Impact of Demographic Characteristics and Chronic Disease with Mortality of COVID-19: Retrospective study

PLOS ONE

Dear Dr. AL MUTAIRI,

Thank you for submitting your manuscript to PLOS ONE. After careful consideration, we feel that it has merit but does not fully meet PLOS ONE’s publication criteria as it currently stands. Therefore, we invite you to submit a revised version of the manuscript that addresses the points raised during the review process.

The manuscript has been assessed by two reviewers. Their comments are appended below. The reviewers have raised some of major concerns about the manuscript, and in particular they feel that some methodological issues exist (especially data analysis) that affect the technical soundness of your study.

We look forward to receiving your revised manuscript.

Kind regards,

Bruno Pereira Nunes, Ph.D.

Academic Editor

PLOS ONE

Journal Requirements:

2. Thank you for including in the text of your manuscript:  "Prior to data collection, an

expedite approval was obtained from Institutional Review Board (IRB Log No: RC.

RC20.09.10). Patients were de-identified and personal information were collected and

therefore no informed consent was deemed necessary. In the present study the ethical

guidelines of Declaration of Helsinki and good clinical practice was followed."

Please add this information to your ethics statement in the online submission form.

In addition, please provide the full name of the ethics committee/institutional review board(s) that approved your specific study to your manuscript text and to your ethics statement in the online submission form.

3. Thank you for providing the date(s) when patient medical information was initially recorded. Please also include the date(s) on which your research team accessed the databases/records to obtain the retrospective data used in your study.

4. We suggest you thoroughly copyedit your manuscript for language usage, spelling, and grammar. If you do not know anyone who can help you do this, you may wish to consider employing a professional scientific editing service.  

6. We note you have included a table to which you do not refer in the text of your manuscript. Please ensure that you refer to Table 1, 3, 4 and 5 in your text; if accepted, production will need this reference to link the reader to the Table.

Reviewers' comments:

Reviewer's Responses to Questions

**Comments to the Author**

1. Is the manuscript technically sound, and do the data support the conclusions?

Reviewer #1: Partly

Reviewer #2: Partly

2. Has the statistical analysis been performed appropriately and rigorously? 

Reviewer #1: No

Reviewer #2: Yes

3. Have the authors made all data underlying the findings in their manuscript fully available?

Reviewer #1: No

Reviewer #2: No

4. Is the manuscript presented in an intelligible fashion and written in standard English?

Reviewer #1: Yes

Reviewer #2: Yes

5. Review Comments to the Author

Reviewer #1: Abstract: The background does not included the mortality situation of COVID-19 in Saudi Arabia. Additionally, it is interesting to mention the place and time period of data collection.

1) In the introduction, I would suggest some references mentioning the situation of covid-19 mortality in Saudi Arabia. Furthermore, citing the relationship between the disease, other comorbidities, and the sociodemographic characteristics.

2) In Methods, correct the sentence: "Data were recruited from April 2020 to June

202, data collection incuded revieweing patients’ electronic medical records, nursing notes, laboratory charecterstics, management details and clinical outcomes.”

3) The authors report that they collected data about “patients’ electronic medical records, nursing notes, laboratory characteristics, management details and clinical outcomes”. Why were not all these data used in the statistical analysis?

4) In the data analysis, I suggest describing the variables compared to the mortality group (yes or no). These comparisons should evaluate using a statistical test. Moreover, indicate which program was performed the statistical analysis.

5) In the Results section, I suggest highlight the most important findings and joining tables 1 and 2. In addiction, I suggest placing all comparisons between mortality groups in this unified table.

6) In Table 3, I would show the logistic regression (crude and adjusted) for the variables in the joined table. I suggest presenting only odds ratio (OR), confidence interval (CI 95%), and p-value.

7) In the Table 5, I would show only odds ratio (OR), confidence interval (CI 95%), and p-value.

8) The Discussion section will need review, including other references. I would suggest better elaborating on the limitations of the study. For instance, I would include the sample size, and other comorbidities.

Reviewer #2: Title - Authors should consider 'impact on' instead of 'impact with'

Methods - Line 2: Authors should take off 'have been employed' , Line 3: replace 'connection' with 'association'

Design - line 3: Authors should check spelling of 'reviewing'. Line 7- Authors should clarify collection of personal data

Data Analysis - Analysis is incomplete to draw conclusion on the association of demographic characteristics with morbidity among COVID-19 patients. Other demographic characteristics though not of interest to authors could also have been included in the study and controlled for, to better illustrate the impact of the demographic characteristics of interest to the authors. eg. educational status, marital status, occupation status etc. Thus, the analysis raises important concerns and questions about what informed the inclusion of only some variables in the analysis?

Table 1 - Why baseline? since there is no end line data presented? Also, authors should consider using ;age(years) and state only numbers in the columns and avoid repeating 'years' in the column in the table in addition to the figures

table 3 -Authors should state the exact p-values. Authors should clarify their level of significance - is it <0.05 or >0.0005? Authors should state p-values to 3 decimal places.

Table 4- Some p-values are stated in exact figures while others are not. mixture does not sound statistically acceptable.

Table 5 -Analysis is not clear, what is the objective of analyzing HPT VS. DM? Do the authors mean HPT & DM instead?

6. PLOS authors have the option to publish the peer review history of their article (what does this mean?). If published, this will include your full peer review and any attached files.

Reviewer #1: No

Reviewer #2: **Yes: **Mary Eyram Ashinyo

---

## [Author Response · Author response to Decision Letter 0]

12 Apr 2021

Thank you once again for the feedback received, and we hope that this information fulfils all of your requirements. Should you have any queries or require further information please do not hesitate to contact me for clarification.

---

## [Decision Letter · Decision Letter 1]

1 Jun 2021

PONE-D-21-01309R1

Examining and Investigating the Impact of Demographic Characteristics and Chronic Disease on Mortality of COVID-19: Retrospective Study

PLOS ONE

Dear Dr. AL MUTAIRI,

Thank you for submitting your manuscript to PLOS ONE. After careful consideration, we feel that it has merit but does not fully meet PLOS ONE’s publication criteria as it currently stands. Therefore, we invite you to submit a revised version of the manuscript that addresses the points raised during the review process.

The manuscript has been assessed by one reviewer. The comments are appended below. The reviewer has raised some concerns about the manuscript. Furthermore, it is recommended to authors present more details about data collection and the ethical statements.

We look forward to receiving your revised manuscript.

Kind regards,

Bruno Pereira Nunes, Ph.D.

Academic Editor

PLOS ONE

Journal Requirements:

Reviewers' comments:

Reviewer's Responses to Questions

**Comments to the Author**

1. If the authors have adequately addressed your comments raised in a previous round of review and you feel that this manuscript is now acceptable for publication, you may indicate that here to bypass the “Comments to the Author” section, enter your conflict of interest statement in the “Confidential to Editor” section, and submit your "Accept" recommendation.

Reviewer #1: All comments have been addressed

2. Is the manuscript technically sound, and do the data support the conclusions?

Reviewer #1: Partly

3. Has the statistical analysis been performed appropriately and rigorously? 

Reviewer #1: Yes

4. Have the authors made all data underlying the findings in their manuscript fully available?

Reviewer #1: Yes

5. Is the manuscript presented in an intelligible fashion and written in standard English?

Reviewer #1: Yes

6. Review Comments to the Author

Reviewer #1: 1) In the Introduction section, correct the sentence: “Globally, as of 6 April

2021, there have been 131,020,967confirmed cases of COVID-19, including 2,850,521

deaths, reported to the World Health Organization.”

2) In table 1 (Results section), I would show totals column and round percent values. The comparisons between ‘survivors’ and ‘non-survivors’ should be evaluate using a statistical test.

3) Indicate which program was performed the statistical analysis.

4) In tables 2, 3 and 4 change “sig.” by “p-value”. Additionally, the column order is Variable name, Odds Ratio (OR), CI 95%, and p-value. If the exact p-value is less than 0.001, it is conventional to state merely ‘p < 0.001’. Constant value is not necessary in the logistic regression. Include “adjusted” term in tables 2 and 3.

5) I do not understand the “@” term in the results comments.

6) The findings of the ‘Age’ and ‘Gender’ variables in the text are different of the Table 2.

7) Change OD by OR in the sentence “The results of logistic regression showed that the history of HTN [OD=1.822,P<0.05] @ CI [1.64,5.21] and DM [OD=4.011,P>0.05] @ CI [1.523,10.561] have significant association with mortality”

8) In the Discussion section, I didn’t find references in the results about the sentence “The binary logistic regression analysis used in our study quantitatively suggests that the COVID-19 patients who have age >50 years, who are males, or who have history of HTN and/or DM may have up to 9-fold increased chances of mortality than those patients who are young, who are females and are non-diabetic and normotensive”. The logistic regression was carried out with a continuous age variable, but the discussion focuses on the findings in older patients.

9) Explain how the selection bias that occurred in this study.

7. PLOS authors have the option to publish the peer review history of their article (what does this mean?). If published, this will include your full peer review and any attached files.

Reviewer #1: No

---

## [Author Response · Author response to Decision Letter 1]

7 Jun 2021

Thank you once again for the feedback received, and I hope that this information fulfils all of your requirements. Should you have any queries or require further information please do not hesitate to contact me for clarification.

---

## [Decision Letter · Decision Letter 2]

23 Jun 2021

PONE-D-21-01309R2

Examining and Investigating the Impact of Demographic Characteristics and Chronic Diseases on Mortality of COVID-19: Retrospective Study

PLOS ONE

Dear Dr. AL MUTAIRI,

Thank you for submitting your manuscript to PLOS ONE. After careful consideration, we feel that it has merit but does not fully meet PLOS ONE’s publication criteria as it currently stands. Therefore, we invite you to submit a revised version of the manuscript that addresses the points raised during the review process.

ACADEMIC EDITOR: Please address the comments raised by reviewers. 

We look forward to receiving your revised manuscript.

Kind regards,

Prasenjit Mitra, MD, MRSB, MIScT, FLS, FACSc, FAACC

Academic Editor

PLOS ONE

Journal Requirements:

Additional Editor Comments (if provided):

Reviewers' comments:

Reviewer's Responses to Questions

**Comments to the Author**

1. If the authors have adequately addressed your comments raised in a previous round of review and you feel that this manuscript is now acceptable for publication, you may indicate that here to bypass the “Comments to the Author” section, enter your conflict of interest statement in the “Confidential to Editor” section, and submit your "Accept" recommendation.

Reviewer #1: All comments have been addressed

2. Is the manuscript technically sound, and do the data support the conclusions?

Reviewer #1: Partly

3. Has the statistical analysis been performed appropriately and rigorously? 

Reviewer #1: Yes

4. Have the authors made all data underlying the findings in their manuscript fully available?

Reviewer #1: Yes

5. Is the manuscript presented in an intelligible fashion and written in standard English?

Reviewer #1: Yes

6. Review Comments to the Author

Reviewer #1: 1) In the Introduction section, correct the sentence: “Another unique prognostic factor in Saudi Arabia is the migrant labor workers who are mostly men coming from less developed countries and known of increasing of increasing prevalence of cardiometabolic diseases.”

2) Include bibliography reference about SPSS program

3) In tables 2, 3 and 4 change “Exp(B)” by “Odds Ratio (OR)”. Additionally, the column order is Variable name, Odds Ratio (OR), 95% CI, and p-value. If the exact p-value is less than 0.001, it is conventional to state merely ‘p < 0.001’. Constant value is not necessary in the logistic regression. Include “adjusted” term in tables 2 and 3.

4) I do not know the ‘@’ term in the confidence interval. I think that the correct term is “OR=2.77 (95% CI 1.60 - 4.78)”, for example. Furthermore, use this sentence for comment the results. So, you do not need to use p-value together with 95% CI.

7. PLOS authors have the option to publish the peer review history of their article (what does this mean?). If published, this will include your full peer review and any attached files.

Reviewer #1: No

---

## [Author Response · Author response to Decision Letter 2]

24 Jun 2021

Thank you once again for the feedback received, and I hope that the information in Rebuttal letter (attached) fulfils all of your requirements. Should you have any queries or require further information please do not hesitate to contact me for clarification.

---

## [Decision Letter · Decision Letter 3]

29 Jul 2021

PONE-D-21-01309R3

Examining and Investigating the Impact of Demographic Characteristics and Chronic Diseases on Mortality of COVID-19: Retrospective Study

PLOS ONE

Dear Dr. AL MUTAIRI,

Thank you for submitting your manuscript to PLOS ONE. After careful consideration, we feel that it has merit but does not fully meet PLOS ONE’s publication criteria as it currently stands. Therefore, we invite you to submit a revised version of the manuscript that addresses the points raised during the review process.

ACADEMIC EDITOR: Please check and revise accordingly 

We look forward to receiving your revised manuscript.

Kind regards,

Prasenjit Mitra, MD, MRSB, MIScT, FLS, FACSc, FAACC

Academic Editor

PLOS ONE

Journal Requirements:

Reviewers' comments:

Reviewer's Responses to Questions

**Comments to the Author**

1. If the authors have adequately addressed your comments raised in a previous round of review and you feel that this manuscript is now acceptable for publication, you may indicate that here to bypass the “Comments to the Author” section, enter your conflict of interest statement in the “Confidential to Editor” section, and submit your "Accept" recommendation.

Reviewer #1: All comments have been addressed

2. Is the manuscript technically sound, and do the data support the conclusions?

Reviewer #1: Yes

3. Has the statistical analysis been performed appropriately and rigorously? 

Reviewer #1: Yes

4. Have the authors made all data underlying the findings in their manuscript fully available?

Reviewer #1: Yes

5. Is the manuscript presented in an intelligible fashion and written in standard English?

Reviewer #1: Yes

6. Review Comments to the Author

Reviewer #1: 1) In the Results section, correct all “OR” and “95%CI” value sentences with three decimal places.

2) In tables 2 and 4 replace the p-values “0.0005” with “< 0.001”.

3) In the Discussion section, I don't think it's interesting to put the term "etc" in scientific articles. Furthermore, in the last paragraph of the discussion section, include the bibliographic references.

7. PLOS authors have the option to publish the peer review history of their article (what does this mean?). If published, this will include your full peer review and any attached files.

Reviewer #1: No

---

## [Author Response · Author response to Decision Letter 3]

29 Jul 2021

Thank you once again for the feedback received, and I hope that corrected paper and our responses letter fulfils all of your requirements. Should you have any queries or require further information please do not hesitate to contact me for clarification.

---

## [Editor Report · Decision Letter 4]

25 Aug 2021

Examining and Investigating the Impact of Demographic Characteristics and Chronic Diseases on Mortality of COVID-19: Retrospective Study

PONE-D-21-01309R4

Dear Dr. AL MUTAIRI,

We’re pleased to inform you that your manuscript has been judged scientifically suitable for publication and will be formally accepted for publication once it meets all outstanding technical requirements.

Kind regards,

Prasenjit Mitra, MD, CBiol, MRSB, MIScT, FLS, FACSc, FAACC

Academic Editor

PLOS ONE
---

## [Editor Report · Acceptance letter]

2 Sep 2021

PONE-D-21-01309R4 

Examining and Investigating the Impact of Demographic Characteristics and Chronic Diseases on Mortality of COVID-19: Retrospective Study 

Dear Dr. Al Mutairi:

I'm pleased to inform you that your manuscript has been deemed suitable for publication in PLOS ONE. Congratulations! Your manuscript is now with our production department. 

Kind regards, 

on behalf of

Dr. Prasenjit Mitra 

Academic Editor

PLOS ONE